# Endothelium-Dependent Hyperpolarization (EDH) in Diabetes: Mechanistic Insights and Therapeutic Implications

**DOI:** 10.3390/ijms20153737

**Published:** 2019-07-31

**Authors:** Kenichi Goto, Takanari Kitazono

**Affiliations:** Department of Medicine and Clinical Science, Graduate School of Medical Sciences, Kyushu University, Fukuoka 812-8582, Japan

**Keywords:** antidiabetic agent, Ca^2+^-activated K^+^ channel, diabetes mellitus, endothelial function, endothelium-dependent hyperpolarization, endothelium-derived hyperpolarizing factor, gap junction, reactive oxygen species

## Abstract

Diabetes mellitus is one of the major risk factors for cardiovascular disease and is an important health issue worldwide. Long-term diabetes causes endothelial dysfunction, which in turn leads to diabetic vascular complications. Endothelium-derived nitric oxide is a major vasodilator in large-size vessels, and the hyperpolarization of vascular smooth muscle cells mediated by the endothelium plays a central role in agonist-mediated and flow-mediated vasodilation in resistance-size vessels. Although the mechanisms underlying diabetic vascular complications are multifactorial and complex, impairment of endothelium-dependent hyperpolarization (EDH) of vascular smooth muscle cells would contribute at least partly to the initiation and progression of microvascular complications of diabetes. In this review, we present the current knowledge about the pathophysiology and underlying mechanisms of impaired EDH in diabetes in animals and humans. We also discuss potential therapeutic approaches aimed at the prevention and restoration of EDH in diabetes.

## 1. Introduction

Endothelial cells play a critical role in the regulation of vascular tone through the release of endothelial-derived relaxing and constricting factors [1]. Nitric oxide (NO) contributes greatly to the endothelium-dependent relaxation in large-conduit arteries, but the hyperpolarization of vascular smooth muscle cells mediated by endothelial cells is the predominant mechanism that explains the endothelium-dependent relaxation in small resistance arteries [1]. Depending on the vascular beds and species, electrical coupling between endothelial cells and smooth muscle cells via myoendothelial gap junctions (MEGJs) and/or endothelium-derived diffusible substances contributes to the endothelium-dependent smooth muscle hyperpolarization [2,3,4,5,6].

Endothelial stimulation with agonists or by shear stress increases the intracellular calcium concentrations, which in turn generates endothelial hyperpolarization through the opening of small (SK_Ca_) and intermediate conductance (IK_Ca_) Ca^2+^-activated K^+^ channels [2,3,4,5,7]. Then, in a number of arteries in which MEGJs exist, the endothelium-dependent hyperpolarization (EDH) spreads to adjacent smooth muscle cells via MEGJs, leading to vasorelaxation [2,3,4,5,8,9]. Although the intracellular Ca^2+^ release from the endoplasmic reticulum (ER) and the subsequent activation of the SK_Ca_ and IK_Ca_ channels is an initial step for the generation of EDH [3,4,7], the Ca^2+^ influx through endothelial nonselective cation channels of the transient receptor potential (TRP) family after ER calcium depletion also contributes to the generation of EDH via the downstream activation of SK_Ca_ and IK_Ca_ channels (Figure 1) [2,3,4,5,10,11,12]. 

In some vascular beds, the rise in the intracellular calcium concentration causes a release of diffusible substance termed endothelium-derived hyperpolarizing factor (EDHF) which are distinct from NO or vasodilator prostanoids. Several factors such as epoxyeicosatrienoic acids (EETs), K^+^ ions, C-type natriuretic peptide (CNP), hydrogen peroxide, and hydrogen sulfide (H_2_S) have been proposed for the nature of EDHF [2,4,6]. Although these diffusible factors in general hyperpolarize the membrane via the activation of smooth muscle potassium channels and/or the Na -K pump, these factors also act on endothelial potassium channels to generate or amplify EDH in certain vascular beds in specific conditions (Figure 1) [13,14].

Diabetes mellitus is a metabolic disease characterized by high levels of blood glucose resulting from defects in insulin secretion and/or insulin action [15]. Long-term diabetes mellitus causes macrovascular and microvascular complications, and endothelial dysfunction appears to play a pathophysiological role in the incidence and development of these complications [15]. Since EDH/EDHF represents a predominant vasodilatory mechanism in small resistance arteries [1,2,3,4], it is plausible to hypothesize that an impairment of EDH/EDHF would particularly contribute to the incidence and progression of diabetic microvascular complications such as retinopathy, nephropathy, and neuropathy. Moreover, impairment of EDH/EDHF in diabetes would increase the peripheral vascular resistance and thus the arterial blood pressure, which could further accelerate the progression of the vascular complications associated with diabetes. In addition to diabetic vascular complications, diabetic cardiomyopathy is also a major cause of mortality and morbidity in patients with diabetes mellitus [16], and EDH/EDHF may have direct effects on cardiomyocytes or modulate diabetic cardiomyopathy through its effects on vascular biology.

The increasing prevalence of diabetic mellitus is a global public health problem [17], and it is thus of clinical importance to elucidate the underlying mechanisms of diabetic microvascular complications and to identify effective treatments. In this review, we summarize the relevant studies in animals and humans, and we address the pathogenesis and possible treatment of impaired EDH/EDHF in diabetes mellitus.

## 2. EDH in Animal Models of Diabetes

In humans, type 1 diabetes is characterized by an autoimmune destruction of the pancreatic β cells, leading to a lack of insulin secretion. Animal models of type 1 diabetes have been created by destroying the pancreatic β cells with streptozotocin (STZ), and most of the studies examining EDH-mediated responses in type 1 diabetes have been investigated using STZ-treated rodents [18]. In 1997, Fukao et al. revealed that in the mesenteric arteries of STZ-induced diabetic rats, acetylcholine (ACh)-induced EDH and relaxation resistant to inhibitors of NO and prostaglandin synthesis (and thus EDH-mediated responses) were reduced [19]. Subsequent studies in STZ-induced diabetic rats and mice also described impaired EDH-mediated responses in mesenteric arteries [20,21,22,23,24,25,26,27], coronary arteries [28,29], retinal arterioles [30], renal microcirculation [31], corpus cavernosum [32], and in arterioles overlying the sciatic nerve [33].

Type 2 diabetes, the most common type of diabetes, is characterized by insulin resistance, inappropriate insulin secretion, and hyperglycemia. Various experimentally induced rodent models of type 2 diabetes have been developed to gain insight into the pathophysiology of human type 2 diabetes [18,34]. These models include the Zucker diabetic fatty (ZDF) rat, the Otsuka Long-Evans Tokushima Fatty (OLETF) rat, the Goto-Kakizaki (GK) rat, and the db/db mouse [18,34]. As with type 1 diabetes, reduced EDH-mediated responses in these rodent models of type 2 diabetes have been reported in a number of vascular beds including mesenteric [35,36,37,38,39,40,41,42], coronary [43], renal [44], cerebral [45], and penile [46] arteries as well as in epineurial arterioles of the sciatic nerve [47].

Thus, in general, EDH-mediated responses are reduced in both type 1 and type 2 animal models of diabetes. However, some studies have reported unaltered [48,49] or even augmented [50,51] EDH-mediated responses in experimental diabetes. The precise reason(s) for the discrepancies among the studies are unclear, but they may be dependent on the severity and/or the duration of diabetes [52]. Alternatively, the unaltered and/or augmented EDH-mediated responses in diabetes could be explained by the theory that EDH is upregulated to maintain overall endothelial function in certain circumstances, in particular when NO-mediated vasorelaxation is compromised [53,54].

Regardless of the underlying mechanisms that lead to the upregulation of EDH in diabetes, such compensatory mechanisms would be expected to disappear when diabetes is sustained over a long period of time. Indeed, it was reported that in the skeletal-muscle microvascular circulation in a primate model of diet-induced obesity and insulin resistance, a compensatory upregulation of EDH was sustained for >18 months after the start of a high-fat diet and then abruptly disappeared at 24 months [55]. Thus, although EDH may be upregulated in some circumstances (particularly in early-stage diabetes), long-term diabetes would produce an impairment of EDH, and this impairment would aggravate the microvascular and macrovascular complications associated with diabetes.

## 3. Mechanisms of Impaired EDH in Diabetes

### 3.1. The Role of Intracellular Ca^2+^
*Mobilization*

The membrane potential changes induced by EDH are typically composed of two phases: An initial rapid phase followed by a sustained second phase [4,19]. The initial rapid phase appears to be provided by the Ca^2+^ released from intracellular stores, and the sustained second phase seems to be due to the Ca^2+^ influx through ion channels located on the cell membrane [56,57]. Thus, dysregulation of these Ca^2+^ signaling pathways in endothelial cells would exert a deleterious effect on EDH-mediated responses.

Changes in intracellular Ca^2+^ mobilization in response to a high glucose concentration have been reported in cultured vascular endothelial cells. An exposure to high glucose enhanced the agonist-stimulated Ca^2+^ mobilization in porcine aortic endothelial cells [58], and subsequent studies showed inhibitory effects of high glucose on endothelial Ca^2+^ mobilization upon agonist stimulation [59,60,61,62] or as a consequence of the accumulation of reactive oxygen species (ROS) [59,60] or excessive protein kinase C (PKC) activation [60,61]. In addition, in cultured endothelial cells from bovine aorta and rat heart, the glycation of extracellular matrix proteins has been shown to impair agonist-stimulated Ca^2+^ mobilization, possibly due to increased oxidative stress [63]. These results may provide a rational explanation for the previous data describing inhibitory effects of high glucose on EDH-mediated responses in a ROS-dependent manner in some vascular beds [64,65,66].

Impaired intracellular Ca^2+^ mobilization upon agonist stimulation has also been reported in both freshly isolated endothelial cells and endothelial cells in an isolated intact arterial segment (native endothelial cells) from diabetic rats and mice [42,67,68,69]. In freshly isolated coronary endothelial cells from STZ-induced diabetic mice, impaired endothelial Ca^2+^ mobilization was due to a decrease in the Ca^2+^ released from the ER [68]. In native aortic endothelial cells from STZ-induced diabetic mice, both the Ca^2+^ release from intracellular stores and the Ca^2+^ influx from the extracellular space were compromised, possibly as a result of increased lysophosphatidylcholine (LPC) released from oxidized low-density lipoprotein (ox-LDL) [67]. Since the plasma concentration of ox-LDL is elevated in STZ-induced diabetic rats [70], and because LPC inhibits EDH-mediated responses in some vascular beds [67,71,72], it is tempting to speculate that LPC released from ox-LDL, at least in part, impairs EDH-mediated responses by decreasing endothelial Ca^2+^ rise in STZ-induced diabetes.

Accumulating evidence suggests that nonselective cation channels of the transient receptor potential (TRP) family in endothelial cells play a crucial role in agonist-stimulated Ca^2+^ influx, which in turn induces endothelium-dependent vasorelaxation in a number of vascular beds [10,11]. In particular, recent studies highlight the pathophysiological role of endothelial TRP vanilloid type 4 (TRPV4) channels in disease-associated endothelial dysfunction [73,74]. In relation to diabetes, high glucose downregulated the protein expression of TRPV4 channels, thereby attenuating the agonist-stimulated Ca^2+^ influx in retinal microvascular endothelial cells [75]. A reduced protein expression of endothelial TRPV4 channels has also been reported in retinal arterioles [75] and mesenteric arteries [24] from STZ-induced diabetic rats, and as such this expression was associated with impaired EDH-mediated responses in mesenteric arteries of this diabetes rat model [24].

In this scenario, a study by Cassuto et al. [76] is highly interesting. They showed that the membrane-localized caveolin-1, a major structural protein of the caveolae [77], was decreased in both high glucose-exposed human coronary endothelial cells and coronary endothelial cells from type 1 and type 2 diabetic patients [76]. Moreover, the number of endothelial caveolae quantified by electron microscopy was significantly decreased in patients with diabetes, possibly due to the disruption of caveolae by peroxynitrite [76]. Taking these results together in conjunction with a recent study that showed the co-localization of TRPV4 channels with caveolin-1 in the caveolae of arterial endothelial cells [78], it is apparent that a decrease in the number of caveolae might underpin the reduced expression of endothelial TRPV4 channels and thus impaired EDH during diabetes in some vascular beds.

### 3.2. The Role of Endothelial Potassium Channels

The rise in the intracellular Ca^2+^ concentration in endothelial cells in turn generates EDH through the downstream activation of SK_Ca_ and IK_Ca_ channels in a number of vascular beds [2,3,4]. In addition, inwardly rectifying (Kir) channels function as an amplifier of EDH in some vascular beds [13,79,80,81]. Thus, changes in the function and/or expression of these potassium channels could also contribute to the altered EDH-mediated responses in diabetes.

In mesenteric arteries of STZ-induced type 1 diabetic rats and mice, reduced responses to K_Ca_ channel activators have been observed [21,23,25]. However, in that vascular bed, controversies exist regarding the expressions of SK_Ca_ and/or IK_Ca_ channels among different studies: Decreased, unchanged, or even increased (Table 1) [22,24,26,27]. In uteroplacental arteries from STZ-induced diabetic pregnant rats, impaired K_Ca_ channel function along with unchanged expressions of SK_Ca_ and IK_Ca_ channel proteins was observed (Table 1) [82]. By contrast, in corpus cavernosum from STZ-induced diabetic rats in which the EDH-mediated relaxation is compromised [32], reduced expressions of SK_Ca_ and IK_Ca_ channel proteins were detected (Table 1) [83]. Thus, although the function of the K_Ca_ channels appears to be impaired, the expressions of SK_Ca_ and/or IK_Ca_ channels have shown variable changes in the vasculature of STZ-induced diabetic rats and mice.

In contrast to the results from STZ-induced type 1 diabetic rodents, in arteries from rodent models of type 2 diabetes, inconsistent results have been observed in studies examining the vasorelaxant responses to K_Ca_ channel activators: Decreased [37,38,39,40,43], unaltered [35,45,84,85], and increased [51,86] (Table 2). The reason for these discrepant results is not clear, but the differences in the duration and/or severity of diabetes may be involved. Indeed, in most but not all cases, unaltered or increased K_Ca_ channel function in type 2 diabetes is associated with a relatively short duration of diabetes (<15 weeks) [45,85] and/or mild hyperglycemia (<10 mmol/L) [45,51,84,86] (Table 2).

Thus, in the vasculature of type 2 diabetic rats and mice, although the K_Ca_ channel function may be preserved or even upregulated at the early-stage and/or mild diabetes, sustained and/or severe diabetes appears to impair the K_Ca_ channel function. However, these functional changes in K_Ca_ channels were not necessarily accompanied by parallel changes in the expression of SK_Ca_ and/or IK_Ca_ channels, as has been observed in STZ-induced type 1 diabetic rats and mice (Table 1 and Table 2). The unaltered or increased expression of K_Ca_ channels may be due to a compensatory upregulation of these channels.

The underlying mechanism that leads to the reduced K_Ca_ channel function during diabetes is not known, but several possibilities can be suggested. One possibility is that compromised endothelial Ca^2+^ mobilization (i.e., a reduction in the intracellular Ca^2+^ release and/or extracellular Ca^2+^ influx) during diabetes indirectly decreases the downstream K_Ca_ channel activation [67,68,69]. Another possibility is that the K_Ca_ channel activity per se is reduced during diabetes: Brøndum et al. showed that the K_Ca_ channel function is reduced in a cytosolic free Ca^2+^-independent manner in mesenteric arteries of Zucker fatty rats (a model of obese and type 2 diabetes) [38]. Together with other studies showing an unaltered or even increased expression of K_Ca_ channels in this vascular bed [35,38,40], the finding by Brøndum et al. may indicate that the reduced K_Ca_ channel activity per se rather than an altered expression of K_Ca_ channels underpins the impairment of the K_Ca_ channel function in this model [38].

The degradation of endothelial glycocalyx — a complex external layer of the endothelial cells that is made up of proteoglycans, glycoproteins, and glycolipids [87] — might also contribute to the impaired EDH-mediated responses in diabetes through a reduction of the SK_Ca_ channel input to EDH [88]. Although speculative, as suggested in the paper by Dogné et al., a thicker glycocalyx might inhibit the downregulation of the SK_Ca_ channel expression on the surface of endothelial cells through preventing access of circulating inflammatory cells to the endothelium [88]. In such cases, when the K_Ca_ channel activity per se is compromised, the direct activation of endothelial K_Ca_ channels by pharmacological modulators of K_Ca_ channels may serve as a promising treatment strategy to ameliorate impaired EDH in diabetes [38,45,89].

Given that diabetes is accompanied by increased ROS production which in turn modulates ion channels in certain vascular beds [90,91,92], it is tempting to hypothesize that reduced K_Ca_ channel activity and/or decreased K_Ca_ channel expression during diabetes are mediated by ROS. Indeed, an inhibition of the K_Ca_ channel activity per se by ROS has been reported in vascular endothelial cells. The IK_Ca_ channel currents recorded by a whole-cell patch clamp in human umbilical vein endothelial cells (HUVECs) and bovine aortic endothelial cells were inhibited by superoxide and hydrogen peroxide, respectively [93,94]. Moreover, ROS may reduce the K_Ca_ channel function via the downregulation of K_Ca_ channel expression in some vascular endothelial cells [93,95]. Advanced glycation end products (AGEs), the formation of which is accelerated during diabetes [96], may promote ROS generation and thus reduce the K_Ca_ channel function by dysregulating the intracellular Ca^2+^ mobilization [97] or by downregulating the expression of K_Ca_ channel proteins [98] in certain vascular endothelial cells.

Downregulation of the SK_Ca_ channel expression by ROS may also lead to arrhythmogenesis in diabetes. In the atria of STZ-induced diabetic mice, increased oxidative stress reduced the expression of SK_Ca_ channel proteins, resulting in action potential prolongation and arrhythmias [99].

In addition to endothelial S/IK_Ca_ channels, endothelial Kir channels also contribute to the generation of EDH in certain vascular beds [13,79]. Intriguingly, some studies reported that the reduced endothelial Kir channel function and expression partly account for impaired EDH in diet-induced obese rats [86,100], and the loss of Kir channels input to EDH in these models might be mediated by a negative influence of hypercholesterolemia on the activity of Kir channels [100,101,102,103]. Since dyslipidemia is a common feature of diabetes, it is worthwhile to investigate the possible involvement of Kir channels in the pathogenesis of diabetic vascular complications.

### 3.3. The Role of Gap Junctions

EDH initiated in endothelial cells spreads to adjacent smooth muscle cells via myoendothelial gap junctions (MEGJs) in many arteries [2,3,4,5]. A gap junction channel is composed of two hemichannels (connexons), and each connexon is comprised of six subunit proteins named connexins (Cx) [104]. It is generally agreed that in rodent and human blood vessels, four proteins (Cx37, Cx40, Cx43, and Cx45) are expressed in the gap junctions [104,105]. Vascular endothelial cells express Cx37, Cx40, and Cx43, and vascular smooth muscle cells express Cx43 and Cx45 [104,105]. With respect to the Cx isoform present at MEGJs, Cx37, Cx40, and to a lesser extent Cx43 have been reported [104]. Since gap junction channels formed by different connexin isoforms have different biophysical properties [104], changes in the number and/or function of connexins that comprise MEGJs could lead to the impaired EDH-mediated responses in diabetes.

In line with this theory, several studies using cell culture methods revealed that high glucose reduced the dye transfer through gap junctions in vascular endothelial and smooth muscle cells because of the phosphorylation of Cx43 via PKC [106,107] or a reduction in Cx43 expression [108,109]. Since dye transfer is thought to occur through gap junctions between electrically coupled cells [110], the reduced function and/or expression of Cx43 could have an impact on EDH-mediated responses in blood vessels. Indeed, the physiological relevance of a high glucose-induced disruption of gap junction activity to EDH-mediated responses has been suggested in experiments using isolated vessels, although the isoform of connexin involved in these studies is not known [111,112]. In retinal microvessels from STZ-induced diabetic rats, an activation of PKC by vascular endothelial growth factor inhibited the electrical transmission along the axis of the vessels; this result might be due to the inhibition of gap junctional communication via PKC [113].

The downregulation of other isoforms of connexin proteins has also been reported in arteries from animal models of type 1 and type 2 diabetes [28,36,114]. Reduced Cx37 and Cx40 protein expression was observed in endothelial cells from coronary arteries of STZ-induced type 1 diabetic mice in which EDH-mediated responses were impaired [28]. In that model, in addition to the reduction in Cx40 expression, a reduced function of Cx40 due to an *O*-linked *N*-acetylglucosaminylation of Cx40 proteins was suggested as an underlying mechanism of impaired EDH-mediated responses [114]. Similarly, in mesenteric arteries of insulin-resistant obese Zucker rats (a model of type 2 diabetes), decreased Cx40 proteins appears to contribute to the impaired EDH-mediated responses [36].

Together these studies suggest that changes in the expression and/or the function of connexins that comprise MEGJs could underlie the impaired EDH-mediated responses in animal models of type 1 and type 2 diabetes. Nevertheless, caution should be taken in generalizing these results because Cx protein expressions and EDH-mediated responses are not necessarily causally related, as was shown in mesenteric arteries of spontaneously hypertensive rats [79,115].

Gap junctional permeability is regulated dynamically by intracellular messengers such as cAMP and cGMP [116]. It was reported that cAMP facilitates EDH-mediated responses by enhancing the electrotonic conduction through both myoendothelial and homocellular smooth muscle gap junctions in some [117] but not all [118,119] vascular beds. Interestingly, Matsumoto et al. demonstrated that impaired EDH-type relaxation is attributable, at least in part, to a reduction in the action of cAMP as a result of both increased phosphodiesterase (PDE) activity and decreased cAMP-dependent protein kinase A (PKA) activity in mesenteric arteries of type 1 and type 2 diabetic rats [39,120]. However, some caution is warranted in interpreting these results [39,120], because a recent study by Moreira et al. suggested that the PDE-3 inhibitor cilostazol (which was used as an enhancer of the activity of cAMP in the studies by Matsumoto et al. [39,120]) ameliorated the age-related impairment of EDH via a reduction in the oxidative stress in rat mesenteric arteries [121]. Moreover, in contrast to the results from rat mesenteric arteries [39,120], the activity of cAMP appears to be preserved in the retinal arterioles of STZ-induced diabetic rats, in which EDH-mediated responses are compromised [30].

### 3.4. The Role of ROS

Reactive oxygen species (ROS) are reactive molecules generated from oxygen metabolism that play crucial roles in vascular function and structure [122]. These ROS include superoxide, hydroxyl radical, hydrogen peroxide, singlet oxygen and peroxynitrite, which are produced as a result of electron transfer reactions [122]. The major sources of ROS in vasculature include the nicotinamide adenine dinucleotide phosphate (NADPH) oxidase, xanthine oxidoreductase, and uncoupled endothelial nitric oxide synthase [122].

A growing body of evidence indicates that ROS plays a crucial role in the development of endothelial dysfunction in diabetes [123]. With respect to the interaction between ROS and EDH, as mentioned in the preceding text, ROS may impair EDH-mediated responses in several ways (e.g., the inhibition of intracellular Ca^2+^ mobilization, the oxidation of LDL, the disruption of caveolae, and the inhibition of function and/or expression of the K_Ca_ channel) in vascular endothelial cells of animal models of diabetes.

Further support for the causative link between ROS and impaired EDH during diabetes comes from a number of studies that showed significant improvements in EDH-mediated responses by antioxidants such as α-lipoic acid [124], red wine polyphenols [125], apocynin [126], ebselen [127], allopurinol [128], and tempol [129] in mesenteric arteries of diabetic rats and mice. In renal arteries from STZ-induced diabetic rats, eugenol (a major constituent of clove oil that has an antioxidant property) improved impaired EDH-mediated relaxation [130].

Nevertheless, it should be stressed that caution must be exercised in making generalizations from those reports. Indeed, several studies have found no beneficial effects of antioxidants on impaired EDH-mediated responses in animal models of type 1 and type 2 diabetes. These antioxidants include superoxide dismutase [19,31], catalase [31], tempol [131], angiotensin receptor blocker [37,131], vitamin C [31], vitamin E [132], and flavonoid [133]. It is not likely that these negative results were due to a lack of the ability to delete ROS, because these antioxidants significantly improved NO-mediated relaxation in these studies [21,131,132,133].

The reason for the above-described contradictory results is not known but might be related to the differences in the agonist used, the vascular bed studied, the nature of ROS generated, or the amount and duration of ROS exposure among the studies. In fact, the effects of ROS on EDH were inconsistent and complex: Decreased [134], unaltered [135,136], and increased [137,138] EDH-mediated responses mediated by ROS have been reported in blood vessels from rats and mice. Although poorly understood, ROS, in particular H_2_O_2_, might augment EDH-mediated responses by potentiating intracellular endothelial Ca^2+^ mobilization [138,139] or by exerting excitatory influences on K_Ca_ channels [140] in some vascular beds in diabetes.

To sum up, although several lines of evidence from animal models of diabetes suggest a link between ROS and reduced EDH, a causal relationship between these two factors during diabetes has not yet been established and merits further investigation.

### 3.5. The Role of Inflammatory Cytokines

Emerging evidence suggests that low-grade inflammation is associated with diabetes-related cardiovascular complications including endothelial dysfunction [141]. Interestingly, some studies showed deleterious effects of pro-inflammatory cytokines on EDH in animal and human vessels. For example, it was shown that the pro-inflammatory cytokine interleukin-1beta (IL-1β) inhibited EDH-mediated responses via a decrease in the expression of cytochrome P450 enzymes in rabbit carotid arteries [142]. Tumor necrosis factor-alpha (TNF-α) attenuated EDH-mediated relaxation in human omental arteries [143]. However, a contradictory result was reported by Wimalasundera et al.: TNF-α inhibited NO-mediated but not EDH-mediated relaxation in rat mesenteric arteries [144].

The effects of pro-inflammatory cytokines on EDH in diabetes are also controversial. In coronary arterioles of type 2 diabetic db/db mice, Park et al. reported that impaired EDH-mediated relaxation was restored by a three-day administration of neutralizing antibody to interleukin (IL)-6, indicating that IL-6 exerts a deleterious influence on EDH in this vascular bed [43], whereas in renal arteries of STZ-induced diabetic rats, a one-month treatment with IL-6 markedly restored impaired EDH-mediated relaxation without altering plasma glucose levels [145].

The reason for these inconsistencies are not known, but one possible and fascinating explanation might be that in the latter study [145], the IL-6 infusion might have inhibited the TNF-α production and thus led to EDH restoration. Indeed, an IL-6 infusion inhibited TNF-α production in humans in vivo [146]. Further investigations are needed to gain insight into the mechanisms whereby pro-inflammatory cytokines influence EDH, and to develop a more in-depth understanding of the interplay between IL-6 and TNF-α with reference to endothelial function.

### 3.6. The Roles of Diffusible Factors

In addition to the mechanisms mentioned above, a reduced production and/or bioavailability of diffusible factors (i.e., EDHFs) were also suggested to contribute to the impaired EDH-mediated responses in diabetes. In porcine coronary arterioles, high-glucose incubation impaired bradykinin-induced, EDH-mediated responses via a reduced production of EETs and reduced CYP activity [66]. Moreover, in mesenteric arteries of type 2 diabetic db/db mice, the inhibition of soluble epoxide hydrolase (sEH) — a ubiquitous enzyme that rapidly hydrolyses EETs to less bioactive dihydroxyeicosatrienoic acids [147] — augmented ACh-induced, EDH-mediated relaxation possibly resulting from elevated EET levels [148]. Similarly, in coronary arteries of obese insulin-resistant mice, the inhibition of sEH enhanced NS309 (a S/IK_Ca_ activator)-induced, EDH-type relaxation [149]. These findings suggest that a reduced production and/or bioavailability of EETs may contribute to the impaired EDH-mediated responses in diabetes in some vascular beds.

H_2_S has also been suggested to contribute to the impaired EDH-mediated responses in diabetes in particular circumstances. In mesenteric arteries of type 2 diabetic db/db mice with hyperhomocysteinemia, a suppressed production of H_2_S by hyperhomocysteinemia was responsible for the impaired EDH-mediated relaxation, because of a reduction in IK_Ca_ input to EDH [150]. In this model, a reduction in the cell-surface expressions of SK_Ca_ and IK_Ca_ channels by homocysteine-induced ER stress might also contribute to the impaired EDH-mediated responses [151]. Since the plasma homocysteine levels were increased in type 2 diabetic patients with nephropathy [152], a reduced contribution of H_2_S to EDH-mediated responses might be of clinical relevance for these patients. In this regard, the increased protein expression of IK_Ca_ observed in diabetic rabbit carotid artery [153] may be a compensatory upregulation to counteract the loss of IK_Ca_ activation due to the reduced blood H_2_S concentration in diabetes [154]. Of interest, H_2_S attenuated myocardial fibrosis in STZ-induced diabetic rats possibly through suppressing oxidative stress and ER stress [155].

Leptin, an adipose tissue hormone, was reported to induce EDH-mediated vasorelaxation [156], and this relaxation was provided at least in part by H_2_S [157]. The reduced production of H_2_S during sustained obesity [158] might contribute to the loss of the leptin-induced, EDH-mediated responses observed in rats with long-term obesity [159].

Hydrogen peroxide (H_2_O_2_) acts as a diffusible EDHF in some vascular beds including coronary arteries [160], and a recent report suggested that an excessive increase in H_2_O_2_ has deleterious effects on coronary microcirculation in vivo in diabetic mice [161]. Thus, in db/db mice, the plasma concentration of H_2_O_2_ was increased, and prolonged exposure to excessive H_2_O_2_ impaired TRP vanilloid-type-1 channels (TRPV1) activity, leading to a reduced TRPV1-dependent modulation of coronary blood flow [161].

The activity of neural endopeptidase (NEP), an endogenous neuropeptide-degrading enzyme, appears to be increased in diabetes [162], which would lead to enhanced CNP degradation. In this context, treatment with the vasopeptidase inhibitor AVE7688 (a simultaneous inhibitor of NEP and angiotensin-converting enzyme activity) improved the reduced ACh-mediated vascular relaxation in epineurial arterioles of STZ-induced diabetic rats, at least in part because of an increase in CNP input to EDH [163]. However, the beneficial effect of the NEP inhibitor on EDH via increased CNP bioavailability might be a drug-specific and/or vascular-specific effect, because no significant differences in EDH-mediated responses were observed with the use of the angiotensin type 1 receptor blocker (ARB) valsartan or that of sacubitril/valsartan (a dual blocker of NEP and the renin angiotensin system [RAS]) in mesenteric arteries of spontaneously hypertensive rats (SHRs) [164].

Changes in the function and/or expression of the large-conductance (BK_Ca_) Ca^2+^-activated K^+^ channels located on vascular smooth muscle cells could also contribute to the impaired EDH-mediated responses in arteries in which diffusible factors induce membrane hyperpolarization through the opening of smooth muscle BK_Ca_ channels. Indeed, in coronary arteries from high-fat diet-induced diabetic mice [165] and type 2 diabetic db/db mice [166], the BK_Ca_ channel function was impaired due to the downregulation of the BK_Ca_ channel β1 subunit expression in response to increased oxidative stress [165,166]. Interestingly, in coronary arteries from diabetic mice, an induction of nuclear factor erythroid-2-related factor-2 (Nrf2), a master regulator of antioxidants, restored the BK_Ca_ channel β1 subunit expression and thus augmented BK_Ca_-mediated vasodilation [165,166], indicating that Nrf2 could be a potential therapeutic target to ameliorate impaired EDH in diabetes in some vascular beds.

### 3.7. Other Factors

Structural and functional changes in vascular smooth muscle cells may also be associated with reduced EDH-mediated responses in obesity and diabetes. It has been reported that diabetes is associated with media hypertrophy in a certain vascular bed [167]. Under such circumstances, propagation of EDH might rapidly dissipate across the media in diabetic arteries. In resistance arteries of diet-induced obese rats, sympathetic nerve-mediated vasoconstriction is augmented [168], which could counteract the vasorelaxant effect of EDH.

Moreover, reduced responsiveness of the smooth muscle cells to hyperpolarization stimuli may contribute to impaired EDH-mediated responses in diabetes. Indeed, the endothelium-independent hyperpolarization and relaxation to levcromakalim, a K_ATP_ channel opener, were impaired in arteries of diabetic rats [37,131,169]. Further, in endothelium-denuded mesenteric arteries of STZ-induced diabetic rats, K^+^-induced vasodilation was attenuated, suggesting that the function of Kir channels and Na^+^/K^+^ ATPase in the smooth muscle may be impaired [20].

## 4. Therapeutic Implications

Insulin and different types of oral antidiabetic drugs are used in the treatment of diabetes. Although these drugs prevent the initiation and progression of diabetic complications mainly through their blood glucose-lowering ability, some of the drugs appear to exert additional glucose-independent beneficial effects on the endothelial function [170]. RAS inhibitors or statins are also widely used for diabetic individuals who have hypertension or dyslipidemia, respectively. In this section, we explore the therapeutic impact of these drugs as well as the impact of exercise on the endothelial function, with a focus on EDH in mainly animal models of diabetes.

### 4.1. Insulin

Chronic treatment with insulin prevents or reverses impaired EDH-mediated responses in arteries from STZ-induced diabetic rats [19,171]. It would be logical to speculate that chronic insulin treatment exerts beneficial effects on EDH by lowering the blood glucose levels in this rodent model. Insulin might also contribute to the restoration of impaired EDH in diabetes independently of its glucose-lowering properties. Indeed, it has been reported that insulin directly generated [172,173] or facilitated [174] EDH in some vascular beds. Interestingly, insulin promoted the production of EETs (a candidate EDHF), which in turn induced vasodilation in human radial artery [175]. However, the effect of insulin on EDH is equivocal; acute incubation with insulin (1 mU/mL) inhibited ACh-induced, EDH-mediated relaxation in rat mesenteric arteries [176]. Such an inhibitory effect of insulin on EDH might underpin the impaired EDH-mediated responses observed in a rat model of insulin resistance [177].

### 4.2. Biguanide (Metformin)

Accumulating evidence suggests that metformin, a biguanide oral hypoglycemic agent, exerts direct (other than glucose-lowering) protective effects on vascular endothelial cells, and a number of preclinical and clinical studies have reported that such direct effects of metformin on endothelial cells contribute to the prevention or reduction of diabetic microangiopathy [178]. In this context, the results of several animal studies suggested that metformin directly ameliorates the impaired EDH-mediated responses associated with diabetes [65,179,180,181].

In their study of mesenteric arteries of OLETF rats, Matsumoto et al. suggested that metformin directly improved EDH-mediated relaxation via suppression of vasoconstrictor prostanoids and oxidative stress [179]. In STZ-induced diabetic spontaneously hypertensive rat aorta, chronic treatment with metformin augmented EDH-mediated relaxation, possibly via an upregulation of the synthesis of H_2_S (a candidate EDHF) independently of glycemic control [180]. In these two studies [179,180], however, the reduction of blood pressure by metformin may also contribute to the improvement of EDH.

Metformin was reported to restore an AGE-mediated downregulation of both SK_Ca_ and IK_Ca_ channel protein expressions, possibly by inhibiting AGE-evoked ROS generation in HUVECs [98]. It thus seems likely that metformin improves the impaired EDH associated with diabetes at least partly through mechanisms independent of its glucose-lowering ability. Although several studies have suggested that metformin exerts beneficial effects on the endothelial function through an activation of AMP-activated protein kinase (AMPK) [178], a recent report by Chen et al. showed that an acute activation of endothelial AMPK inhibited EDH-mediated relaxation in rat mesenteric arteries [182].

### 4.3. Dpp-4 Inhibitors And Glp-1r Agonists

Although a number of studies demonstrated direct actions of dipeptidyl peptidase-4 (DPP-4) inhibitors or glucagon-like peptide-1 receptor (GLP-1R) agonists on vascular endothelial cells [183,184], few studies have investigated the possible involvement of EDH in the direct actions of those drugs on vascular endothelium and its alterations in diabetes. In mouse aorta, acute treatment with alogliptin, a DPP-4 inhibitor, induced EDH-mediated relaxation apart from the activation of GLP-1R [185]. In their study of rat mesenteric arteries, Salheen et al. showed that acute treatment with linagliptin (a DPP-4 inhibitor) or extendin-4 (a GLP-1R agonist) prevented the high glucose-induced impairment of EDH through direct ROS scavenging and GLP-1R activation [186].

Salheen et al. reported that chronic treatment with linagliptin improved the EDH-mediated relaxation without decreasing the plasma glucose in mesenteric arteries of STZ-induced diabetic rats, and that the improvement of EDH by linagliptin appears to be due to the suppression of ROS generation [187]. Another DPP-4 inhibitor, vildagliptin, also improved EDH-mediated relaxation independently of glycemic control in STZ-induced diabetic spontaneously hypertensive rat aorta [180].

A direct vasodilatory influence of GLP-1 and its analogues mediated by EDH has also been reported [186,188,189]. Thus, acute treatment with GLP-1(7-36) or its metabolite GLP-1(9-36) induced EDH-mediated relaxation in the third branches of rat mesenteric arteries [188]. Interestingly, in that study, both the GLP-1(7-36)-evoked and GLP-1(9-36)-evoked EDH-mediated relaxations were attenuated in STZ-induced diabetic rats compared to normoglycemic controls [188]. Further, a recent report by Sukumaran et al. showed that chronic treatment with liraglutide, a human GLP-1 analogue, ameliorated the in vivo renal microcirculation of obese Zucker rats fed a high-salt diet, probably due to the enhanced contribution of NO and/or EDH [189].

Collectively, the findings from these studies suggest that both DPP-4 inhibitors and GLP-1R agonists exert beneficial actions on EDH in diabetes through mechanisms independent of their glucose-lowering effects. The underlying mechanisms of such improvements remain unclear and warrant further investigation.

### 4.4. SGLT2 Inhibitors

Emerging evidence suggests that sodium glucose co-transporter2 (SGLT2) inhibitors provide beneficial effects against cardiovascular events beyond their glucose-lowering properties, but the underlying mechanisms of such benefits are not well understood [190]. Acute [191] or chronic [192] treatment with SGLT2 inhibitors was shown to enhance endothelium-dependent vasorelaxation independently of the inhibitors’ glucose-lowering effects, but the contribution of EDH to the restoration of the endothelial function following the SGLT2 inhibitor treatments was not examined [191,192].

Interestingly, a recent report showed that the SGLT2 inhibitor empagliflozin restored the integrity of the endothelial glycocalyx in human abdominal aortic endothelial cells [193]. Since the degradation of endothelial glycocalyx seems to contribute to the impaired EDH-mediated responses in diabetes through a reduction in the SK_Ca_ channel input to EDH [88], it is tempting to speculate that SGLT2 inhibitors ameliorate impaired EDH in diabetes by restoring the integrity of the endothelial glycocalyx.

### 4.5. Renin Angiotensin System Inhibitors

The tissue RAS appears to be involved in pathological mechanisms that lead to diabetic vascular complications [194], and several research groups have investigated the effects of RAS inhibitors on EDH in diabetic rats and mice [37,131,195,196]. In mesenteric arteries of GK rats (models of type 2 diabetes), chronic treatment with the ARB losartan ameliorated impaired EDH-mediated relaxation by enhancing K_Ca_ channel activities [195]. By contrast, in mesenteric arteries of GK rats, chronic treatment with another ARB, candesartan, or with the combination of candesartan and the superoxide dismutase mimetic tempol (a scavenger of both intracellular and extracellular superoxides [131]) did not improve EDH or EDH-mediated relaxation [37,131].

It seems unlikely that such disparities among study results arose from the insufficient dose of candesartan used in the study by Oniki et al. [37,131], because chronic treatment with similar doses of candesartan improved the reduced EDH-mediated responses in the same vascular bed during hypertension and aging [197,198]. In mesenteric arteries of diabetic apolipoprotein E-deficient mice, the combination of the ARB olmesartan and the calcium-channel blocker azelnidipine but not olmesartan alone improved EDH and EDH-mediated relaxation [196]. The mechanisms underlying such improvement remain to be clarified.

The effects of RAS inhibitors on EDH in diabetes are thus equivocal. It is nevertheless important to determine whether RAS inhibitors can ameliorate the impaired EDH associated with diabetes, because the activation of the vascular-tissue RAS induces vascular injury and inflammation, thereby contributing to the development and progression of vascular disease [199].

### 4.6. Statins

Direct (pleiotrophic) beneficial effects of statins on the endothelial function beyond their cholesterol-lowering ability have been firmly established in both animals and humans [200]. However, the literature focusing on the effects of statins on EDH in diabetes is still limited, and the results are inconsistent. In mesenteric arteries of STZ-induced diabetic rats, chronic treatment with rosuvastatin corrected the decreased EDH-mediated relaxation without affecting the plasma cholesterol level [201]. By contrast, no change was observed in EDH after chronic treatment with pravastatin in coronary arteries from OLFTE rats at the early stage of diabetes [49]. Presumably, the disparity between the two studies’ findings [49,201] appears to be due to the differences in the severity of diabetes and/or the timing of the treatment initiation.

The pleiotropic effects of statins on EDH may not be universal. Indeed, in mesenteric arteries of stroke-prone spontaneously hypertensive rats (SHRSP), fluvastatin improved the impaired endothelium-dependent relaxation via a restoration of NO-mediated relaxation without any changes in EDH or EDH-mediated relaxation [202].

### 4.7. Protein Kinase C Inhibitors

PKC activity is enhanced in diabetes, leading to vascular dysfunction in several ways [203]. As noted above, PKC appears to contribute to the high glucose-induced impairment of EDH-mediated responses via the inhibition of both endothelial Ca^2+^ mobilization and gap junctional communication [61,62,106,107]. In this regard, one study investigated the effect of a PKC inhibitor on impaired EDH in diabetes; chronic treatment with LY333531, a specific inhibitor of the PKC β isoform, partially restored the impaired EDH-mediated relaxation in mesenteric arteries of STZ-induced diabetic rats [204].

The generation of thromboxane A_2_ (TXA_2_) is increased in diabetes at least partly due to the enhanced activity of PKC [203]. Since the ACh-induced production of TXA_2_ was increased in mesenteric arteries of OLETF rats [205], and because a TXA_2_ analogue depolarized the membrane potential in rat mesenteric arteries [206], it can be speculated that ACh-induced EDH is opposed by a simultaneous depolarization evoked by TXA_2_ in mesenteric arteries of OLETF rats. Indeed, an interplay between EDH and simultaneous depolarization was reported in mesenteric arteries from SHRs [79,207]. Such interactions between hyperpolarization and depolarization might explain the observation that chronic treatment with a TXA_2_ inhibitor, ozagrel, partially ameliorated the impaired ACh-induced, EDH-mediated relaxation in mesenteric arteries of OLETF rats [205].

### 4.8. Aldose Reductase Inhibitors

An aldose reductase inhibitor (ARI) acts to block the first step of the polyol pathway, which converts glucose to sorbitol with NADPH as a coenzyme [208]. In addition to its protective effect on diabetic neuropathy by suppressing sorbitol and fructose accumulation in nervous tissues, emerging evidence suggests that ARI reduces diabetes complications through its antioxidant as well as anti-inflammatory properties [208]. However, there are only a small number of studies regarding the effects of ARI on EDH in diabetes.

In the mesenteric arteries of STZ-induced diabetic rats, chronic treatment with an ARI, WAY121509, partially restored impaired EDH-mediated relaxation [209]. In accord with that report, chronic treatment with another ARI, minalrestat, ameliorated the EDH-mediated vasodilation in vivo in mesenteric arteries of alloxan-induced diabetic rats [210]. The underlying mechanisms of these improvements are not known.

### 4.9. Exercise

Regular physical exercise is recommended as a non-pharmacological treatment of diabetes, and several studies have described beneficial effects of regular exercise on the endothelial function in animal models of diabetes [211]. Although most of those studies focused on the role of NO, an investigation by Minami et al. demonstrated that exercise training improves impaired EDH-mediated relaxation in OLETF rats (a model of type 2 diabetes), probably by ameliorating hyperglycemia and insulin resistance [212].

Since exercise training decreased the serum concentrations of proinflammatory cytokines such as TNF-α and IL-6 in diabetic rats [213] and because these cytokines inhibited EDH in certain vascular beds [43,142], the decrease in these cytokines might also have contributed to the beneficial effect of exercise on EDH-mediated responses revealed by the Minami et al. study [212].

## 5. EDH in Human Diabetes

Although the contribution of EDH to the regulation of vascular tone has been investigated in several human arteries [214,215], few studies have focused on the role of EDH in human diabetes. In respect to the effects of high glucose on EDH in human arteries, MacKenzie et al. showed that while exposure to high glucose (20 mM, 2 h) inhibited bradykinin-induced, EDH-mediated relaxation in subcutaneous arteries, exposure to high glucose (20 mM, 2 h) augmented bradykinin-induced, EDH-mediated relaxation in mesenteric arteries [216]. Thus, the effects of high glucose on EDH-mediated responses differed depending on the vascular bed examined [216], and the inconsistent results might reflect heterogeneity of EDH/EDHF among the agonists and vascular beds studied [4,26].

With respect to alterations in EDH in human diabetes, several studies have shown impaired EDH-mediated responses. In an examination of human penile resistance arteries, EDH-mediated relaxation was impaired in subjects with type 1 and type 2 diabetes, and the impairment of EDH was restored by acute treatment with calcium dobesilate, an antioxidant and an inhibitor of aldose reductase [217]. Moreover, in human coronary arterioles, NS309 (a S/IK_Ca_ activator)-induced, EDH-type relaxation was impaired in subjects with type 2 diabetes because of the decreased SK_Ca_ and IK_Ca_ channels activity per se [218,219]. In a study of human cutaneous microcirculation in subjects with type 1 diabetes, post-occlusive hyperemia (an index of endothelium-dependent vasodilation) was reduced partially by a decreased contribution of EDH [220]. Finally, a recent report by Duflot et al. demonstrated that flow-mediated endothelium-dependent vasodilatation of the radial artery was impaired in subjects with type 2 diabetes independently of their hypertensive status [175]. Of interest, Duflot et al. revealed that a decreased production of EET (a candidate EDHF) and increased EET degradation by sEH in conjunction with decreased NO bioavailability by ROS were mechanistically involved in the impairment [175].

By contrast, EDH-mediated relaxation was augmented to compensate for reduced NO-mediated relaxation in subcutaneous arteries from individuals with diabetes, and such an augmentation of EDH appeared not to be attributable to the drugs used in that study [221]. The reason(s) for these disparities among study results (i.e., reduced or augmented EDH) are not known, but they might be related to the differences in the duration or the severity of diabetes among the study subjects.

## 6. Conclusions

EDH and EDH-mediated relaxation are impaired in long-term diabetes. Evidence from numerous studies using animal models of diabetes suggests that multifactorial mechanisms contribute to the impaired EDH associated with diabetes. The compromised Ca^2+^ handling in endothelial cells, the reduced function and expression of endothelial ion channels, the disruption of gap junctional communication or the breakdown of caveolae and glycocalyx independently or in combination appear to play a causative role in the impaired EDH in diabetes in a number of vascular beds. A reduced production and/or bioavailability of diffusible factors may also contribute to the impairment of EDH in diabetes in some vascular beds. Several animal studies suggest a causative link between ROS and the diabetes-associated impairment of EDH, but conflicting results showing no detrimental effects of ROS on EDH in diabetes are also reported. Rigorous further investigations are needed to draw a definite conclusion on the interplay between ROS and EDH in diabetes.

Although glucose lowering per se improves reduced EDH in diabetes, some pharmacological drugs appear to exert beneficial effects on EDH independently of their glucose-lowering ability. The extent of the improvement in EDH achieved by pharmacological drug therapy is limited in most studies, and the mechanisms that mediate such improvements are not yet known.

EDH-mediated responses are decreased in some but not all arteries of individuals with type 1 or type 2 diabetes. Given that endothelial dysfunction is implicated in the pathogenesis of vascular complications in diabetes and that EDH plays a pivotal role in the endothelial function in resistance arteries, further explorations of the underlying mechanisms of impaired EDH in diabetes could open new doors for the prevention and treatment of microvascular complications in individuals with diabetes mellitus.

## Figures and Tables

**Figure 1 ijms-20-03737-f001:**
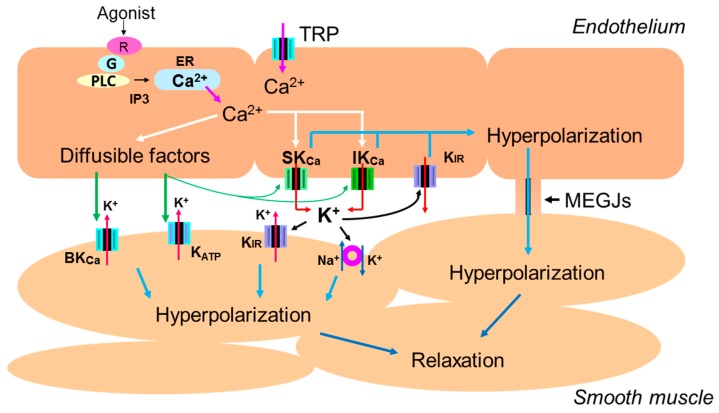
Endothelium-dependent hyperpolarization of vascular smooth muscle cells. Endothelial stimulation with agonists or by shear stress increases the intracellular Ca^2+^ concentration due to Ca^2+^ release from the endoplasmic reticulum (ER) and Ca^2+^ influx through endothelial nonselective cation channels of the transient receptor potential (TRP) family. The rise in the endothelial Ca^2+^ concentration subsequently activates small (SK_Ca_) and intermediate conductance (IK_Ca_) Ca^2+^-activated K^+^ channels, generating endothelium-dependent hyperpolarization (EDH). The EDH then spreads to adjacent smooth muscle cells via myoendothelial gap junctions (MEGJs), leading to vasorelaxation in a number of vascular beds. In some vascular beds, diffusible factors hyperpolarize vascular smooth muscle cells via the opening of potassium channels and/or activation of Na^+^/K^+^-ATPase. Diffusible factors also act on endothelial potassium channels to generate or amplify EDH in certain vascular beds in specific conditions.

**Table 1 ijms-20-03737-t001:** Changes in function and expression of K_Ca_ channels in type 1 diabetes.

Species	Model	Duration of DM	Glucose (mmol/L)	Vascular Bed	Function EDH	Function K_Ca_	Expression SK_Ca_ IK_Ca_		Ref.
Rat	STZ	8 w	31	mesenteric	↓	↓ 1-EBIO	ND	ND	[20]
Rat	STZ	10 w	>33	mesenteric	↓	ND	↑	↑	[21]
Rat	STZ	4 w	24	mesenteric	↓	↓ NS309	ND	ND	[22]
Rat	STZ	12−15 w	>15	mesenteric	↓	ND	↓	ND	[23]
Rat	STZ	12 w	21	mesenteric	↑	ND	ND	ND	[49]
Rat	STZ	18 day	21	uteroplacental	ND	↓ NS309	→	→	[81]
Rat	STZ	8 w	22	corpus cavernosum	↓	ND	↓	↓	[31,82]
Mice	STZ+ApoE^−/−^	10 w	32	mesenteric	↓	↓ NS1619	ND	ND	[24]
Mice	STZ+ApoE^−/−^	12–16 w	>20	mesenteric	↓	ND	↓	→	[25]
Mice	STZ	10 w	44	mesenteric	↓	ND	→	↑	[26]

DM, diabetes mellitus; EDH, endothelium-dependent hyperpolarization; ND, not determined; STZ, streptozotocin; ApoE, apolipoprotein E; ↑, increased; ↓, decreased; →, unchanged.

**Table 2 ijms-20-03737-t002:** Changes in function and expression of K_Ca_ channels in type 2 diabetes.

Species	Model	Duration of DM	Glucose (mmol/L)	Vascular Bed	Function EDH	Function K_Ca_	Expression SK_Ca_ IK_Ca_		Ref.
Rat	ZDF	17–20 w	38	mesenteric	↓	→ 1-EBIO	↑	→	[34]
Rat	ZDF	21 w	24	mesenteric	↓	↓ NS309	→	ND	[37]
Rat	ZDF	18 w	21	mesenteric	↓	↓ 1-EBIO	→	↑	[39]
Rat	ZDF	12–14 w	ND	mesenteric	ND	→ 1-EBIO	ND	↓	[84]
Rat	OZ	20 w	32	renal	↓	↓ NS1619	ND	ND	[43]
Rat	OZ	7–10 w	8.4	cerebral	↓	→ NS309	ND	→	[44]
Rat	OZ	17–18 w	9.1	coronary	ND	↑ NS309	↑	↑	[50]
Rat	OLETF	60 w	19	mesenteric	↓	↓ 1-EBIO	ND	ND	[38]
Rat	OLETF	50–53 w	8.4	mesenteric	↓	↓ NS309	ND	ND	[40]
Rat	Diet	16–20 w	9.8	saphenous	ND	→ 1-EBIO	→	↑	[83]
Rat	Diet	16–20 w	9.7	mesenteric	↓	↑ 1-EBIO	ND	↑	[85]

DM, diabetes mellitus; EDH, endothelium-dependent hyperpolarization; ND, not determined; ZDF, Zucker diabetic fatty; OZ, obese Zucker; OLETF; Otsuka long-evans tokushima fatty; ↑, increased; ↓, decreased; →, unchanged.

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
