# Peer review of "Endothelium-Dependent Hyperpolarization (EDH) in Diabetes: Mechanistic Insights and Therapeutic Implications"

_ijms, 2019, doi:10.3390/ijms20153737_

Round 1

Reviewer 1 Report

This manuscript is a review for the various aspects of EDH mechanisms and their changes in diabetes model animals and human samples. It is very comprehensively written and provide helpful summaries of the relevant topics. I generally agree to publish the manuscript except a few points.

Kir channels in vascular smooth muscle is a component of EDH phenomenon because the are facilitated by K+ released at the abluminal side of endothelium. However, the authors did not cover the change(s) of Kir in smooth muscle. Only BKCa channels in smooth muscle were briefly mentioned.

p2, line 28-30: It is written that EDH change result in retinopathy or nephropathy. However, since EDH is basically electrical phenomenon, one cannot easily understand why EDH impairment induces retinopathy or nephropathy, i.e. diabetic microvascular complications. Does it imply excessive contraction?

p6. line 20-23 (ref 87); the specific mechanisms associated with glycocalys and EDH change is lacking. Please provide further explanation. 

Author Response

Thank you for the detailed review of our manuscript. We appreciate the comments of the reviewer and have attempted to answer each of the question raised. We feel the comments have helped us significantly improve the paper.

Kir channels in vascular smooth muscle is a component of EDH phenomenon because the are facilitated by K+ released at the abluminal side of endothelium. However, the authors did not cover the change(s) of Kir in smooth muscle. Only BKCa channels in smooth muscle were briefly mentioned.

Thank you for your accurate comment. In accordance with the reviewer’s suggestion, we have added the following sentences in the text (p.10, line 44-49): “Moreover, reduced responsiveness of the smooth muscle cells to hyperpolarization stimuli may contribute to impaired EDH-mediated responses in diabetes. Indeed, the endothelium-independent hyperpolarization and relaxation to levcromakalim, an KATP channel opener, were impaired in mesenteric and cerebral arteries of diabetic rats [37,131,169]. Further, in endothelium-denuded mesenteric arteries of STZ-induced diabetic rats, K+-induced vasodilation was attenuated, suggesting that the function of Kir channels and Na+/K+ ATPase in the smooth muscle may be impaired [20].”

p2, line 28-30: It is written that EDH change result in retinopathy or nephropathy. However, since EDH is basically electrical phenomenon, one cannot easily understand why EDH impairment induces retinopathy or nephropathy, i.e. diabetic microvascular complications. Does it imply excessive contraction?

Thank you for this insightful comment. Because EDH contributes greatly to resistance sized arteries, it is expected that impairment of EDH could lead to vascular smooth muscle contraction as the reviewer has pointed out. Excessive smooth muscle contraction would cause a decrease in blood flow to organs. Eventually, the decrease in blood flow may damage organs such as kidney, retina and neuron. In addition, some diffusible EDHF have anti-inflammatory (e.g., EET [Node K et al, Science 1999]) or anti-fibrotic effect (e.g., H2S [Ref 155]). Thus, impairment in the release of EDHF could exert deleterious effects on organs as well. We have therefore replaced “EDH” with “EDH/EDHF” in the last two paragraphs of the introduction (p.2, line 27- p.3, line 8).

p6. line 20-23 (ref 87); the specific mechanisms associated with glycocalys and EDH change is lacking. Please provide further explanation. 

We appreciate the reviewer for pointing it out. Unfortunately, the specific mechanism by which degradation of endothelial glycocalyx lead to the loss of SKCa in diabetes was not determined in the study by Dogne et al. (Ref. 88). Although speculative, as suggested in the paper by Dogne et al. (Ref. 88), a thicker glycocalyx might inhibit the downregulation of SKCa channel expression on the surface of endothelial cells through preventing access of circulating inflammatory cells to the endothelium. We have included this sentence in the text (p.6, lines 23-25).

Reviewer 2 Report

The comments:

The manuscript is a very good review in endothelium-dependent hyperpolarization (EDH) and diabetes field. The authors present an update in the role of endothelium-dependent hyperpolarization (EDH) in diabetes and highlight mechanistic insights and therapeutic implications.

This manuscript makes a rigorous and comprehensive presentation of existing data in specialized scientific literature on pathophysiology and underlying mechanisms impaired EDH in diabetes in animals and humans.

The manuscript is very well organized, wrote and the data are well and clear presented. The Figure and Tables are also very well designed and suggestive.

Moreover, the manuscript is state-of-the-art, comprehensive and convincing, therefore in this context my recommendation is accept for publication.

Author Response

# Reviewer 2

We highly appreciate the reviewer’s positive comments. Respectfully, we have revised the manuscript according to the comments of the reviewers. Changes are highlighted in yellow. Thank you very much again for your consideration.

Reviewer 3 Report

This review article summarized the update knowledge on the role of EDH in DM. This provided useful information and well address the critical issues linked to DM vascular lesions and EDH. It will be appreciated if the authors can do adequate response for this reviewer’s concern.

Although EDH plays a role in vascular biology in DM, it also merits to extent this background to DM cardiomyopathy (line 30) since cardiac co-morbidity is believed to be the main issues of DM patients (eg. Lee TI, et al., Diabetes Res Clin Pract. 2013, or Lee TW, et al., Nutr Res. 2015). It is possible that EDH may have direct effects on cardiomyocytes or modulate DM cardiomyopathy through its effects on vascular biology. Please make comments on this point.

DM is highly associated with arrhythmia, especially atrial fibrillation. Does EDH play a role in AF occurrence since AF most occur from pulmonary vein cardiomyocytes, which are regulated by pulmonary vein vascular tone? NO has been shown to reduce pulmonary vein arrhythmogenesis.

The abbreviation (ND, EDF, OZ…) or arrows should be explained in the table 1 and 2 to avoid misunderstanding.

Author Response

# Reviewer 3

Thank you for inviting a revision of our manuscript. We appreciate the comments of the reviewer and have made a point by point response to each, with alterations to the text as indicated. We feel the comments have helped us significantly improve the paper.

Although EDH plays a role in vascular biology in DM, it also merits to extent this background to DM cardiomyopathy (line 30) since cardiac co-morbidity is believed to be the main issues of DM patients (eg. Lee TI, et al., Diabetes Res Clin Pract. 2013, or Lee TW, et al., Nutr Res. 2015). It is possible that EDH may have direct effects on cardiomyocytes or modulate DM cardiomyopathy through its effects on vascular biology. Please make comments on this point.

As suggested, additions have been made to Introduction (p.2, line 32- p.3, line 3): In addition to diabetic vascular complications, diabetic cardiomyopathy is also a major cause of mortality and morbidity in patients with diabetes mellitus [16], and EDH/EDHF may have direct effects on cardiomyocytes or modulate diabetic cardiomyopathy through its effects on vascular biology. In addition, brief addition has been made in the text with respect to the role of H2S (p.10, lines 2-3): “Of interest, H2S attenuated myocardial fibrosis in STZ-induced diabetic rats possibly through suppressing both oxidative stress and ER stress [155].”

DM is highly associated with arrhythmia, especially atrial fibrillation. Does EDH play a role in AF occurrence since AF most occur from pulmonary vein cardiomyocytes, which are regulated by pulmonary vein vascular tone? NO has been shown to reduce pulmonary vein arrhythmogenesis.

We appreciate the pertinent comment of the reviewer. As the reviewer has pointed out, downregulation of SKCa channel may play a role in AF occurrence in diabetes. In complying with the reviewer’s comment, we have added the following sentences (p.7, lines 9-11): “Downregulation of SKCa channel expression by ROS may also lead to arrhythmogenesis in diabetes. In the atria of STZ-induced diabetic mice, increased oxidative stress reduced the expression of SKCa channel proteins, resulting in action potential prolongation and arrhythmias [99].

The abbreviation (ND, EDF, OZ…) or arrows should be explained in the table 1 and 2 to avoid misunderstanding.

As suggested, the abbreviation or arrows has now been explained in the table 1 and 2 to avoid misunderstanding.

Reviewer 4 Report

Manuscript # 553824

Goto & Kitazono prepared a comprehensive review on how endothelium-dependent hyperpolarization (EDH) of resistance arteries is impacted by diabetes (Type 1 & Type 2). In brief, this is an outstanding review as it is a clear development from the fundamental working model of EDH (see Figure in manuscript) to vast, complex speculation on various alterations in SK/IK function (see Tables) and other primary components of EDH (e.g., gap junction connexins) throughout study models of the severity of disease. They also discuss novel roles of reactive oxygen species, inflammatory cytokines (e.g., interleukin-6), and diffusible factors (e.g., hydrogen sulfide) with a complete follow up on therapeutic options (e.g., SGLT2 inhibitors, exercise) and translation to human subjects. It was an honor to review this manuscript while enjoying its highly informative contents.

I only have two general comments as indicated below. As to not influence the authors’ choice of citation record towards certain groups/labs, there are statements below that are not accompanied by specific references but have a substantial representation in the literature nonetheless within the general field of cardiovascular aging and chronic disease.    

1. Section 2, Pg. 3, Lines 26-32; Tables 1 & 2.

The authors correctly pointed to the discrepant findings for changes in SK/IK function with disease across studies & study models. Perhaps the simplified explanation for this as indicated (Section 2, Lines 33-40; Table 1, Pg. 5, Lines 9-16) is satisfactory for readers but there are several other factors to consider that were not covered in this review. As examples, there may be enhanced smooth muscle adrenergic receptor activity, structural remodeling of smooth muscle cells (e.g., hypertrophy), and/or a reduction in myoendothelial coupling (briefly discussed in Section 3.3). Due to such factors, an otherwise normal or augmented EDH may appear diminished during vasodilation because of the loss of smooth muscle compliance per unit of vasodilator signal produced by the endothelium. If the authors feel that there is enough information in this regard to add for diabetes, it is recommended to do so. If not, perhaps the authors should at least mention that such factors exist and could complicate interpretation for the function of EDH during disease.

2. Section 3.4 (Pg. 8, Lines 12-17); Section 3.6 (Pg. 9, Lines 39-44)

The authors discuss the general role of reactive oxygen species (ROS) and how they may decrease KCa channel function. Diabetes may be an exception in this regard (Section 3.6, Lines 39-44) but there is evidence that ROS, particularly in the form of hydrogen peroxide/hydroxyl radicals, may actually increase calcium mobilization through transient receptor potential (TRP) channels and thereby augment KCa function. Although poorly understood, it is possible that direct oxidation of KCa channels [S/I/BK] may increase their probability of opening of the channels as well. I sense that the authors have a strong sense that the variety of interpretation may be dependent on the type and concentration of ROS here as well. Again, if the authors feel that such avenues of alternative interpretation (e.g., how/when ROS can actually augment EDH) have enough of a presence in the diabetes literature, it is recommended that this be discussed as well.

Also, in section 3.6, the authors discuss hydrogen peroxide as a diffusible factor but it also plays a primary role in the ROS signaling pathway with relevance to Section 3.4. Perhaps the authors could briefly mention the sources, components, and general chemistry of the ROS signaling pathway at the beginning of Section 3.4.

Author Response

# Reviewer 4

We wish to express our appreciation to the reviewer for his or her efforts to improve the presentation and understanding of our data. The comments were valuable and we feel the comments have helped us significantly improve the paper.

If the authors feel that there is enough information in this regard to add for diabetes, it is recommended to do so. If not, perhaps the authors should at least mention that such factors exist and could complicate interpretation for the function of EDH during disease.

We appreciate the comment of the reviewer. As suggested, we have added a new section (Section 3.7 Other factors) to provide information on the structural and functional changes in vascular smooth muscle cells in diabetes (p.10, lines 37-49):

“Structural and functional changes in vascular smooth muscle cells may also be associated with reduced EDH-mediated responses in obesity and diabetes. It has been reported that diabetes is associated with media hypertrophy in certain vascular bed [167]. Under such circumstances, propagation of EDH might rapidly dissipates across the media in diabetic arteries. In resistance arteries of diet-induced obese rats, sympathetic nerve-mediated vasoconstriction is augmented [168], which could counteract vasorelaxant effect of EDH.

  Moreover, reduced responsiveness of the smooth muscle cells to hyperpolarization stimuli may contribute to impaired EDH-mediated responses in diabetes. Indeed, the endothelium-independent hyperpolarization and relaxation to levcromakalim, an KATP channel opener, were impaired in arteries of diabetic rats [37,131,169]. Further, in endothelium-denuded mesenteric arteries of STZ-induced diabetic rats, K+-induced vasodilation was attenuated, suggesting that the function of Kir channels and Na+/K+ ATPase in the smooth muscle may be impaired [20].”

Again, if the authors feel that such avenues of alternative interpretation (e.g., how/when ROS can actually augment EDH) have enough of a presence in the diabetes literature, it is recommended that this be discussed as well.

As suggested, the information (how/when ROS can actually augment EDH) has now been briefly added (p.9, lines 1-3): “Although poorly understood, ROS, in particular H2O2, might augment EDH-mediated responses by potentiating intracellular endothelial Ca2+ mobilization [138,139] or by exerting excitatory influences on KCa channels [140] in some vascular beds in diabetes.”

Perhaps the authors could briefly mention the sources, components, and general chemistry of the ROS signaling pathway at the beginning of Section 3.4

As suggested, the sources, components, and general chemistry of the ROS signaling pathway has now been briefly mentioned at the beginning of Section 3.4 (p.8, lines 22-27): “Reactive oxygen species (ROS) are reactive molecules generated from oxygen metabolism that play crucial roles in vascular function and structure [122]. These ROS include superoxide, hydroxyl radical, hydrogen peroxide, singlet oxygen and peroxynitrite, which are produced as a result of electron transfer reactions [122]. The major sources of ROS in vasculature include nicotinamide adenine dinucleotide phosphate (NADPH) oxidase, xanthine oxidoreductase, and uncoupled endothelial nitric oxide synthase [122].”